# Alterations in RNA Expression Profile Following *S. aureus* and *S. epidermidis* Inoculation into Platelet Concentrates

**DOI:** 10.3390/ijms26073009

**Published:** 2025-03-26

**Authors:** Jae Kwon Kim, Taewon Kang, Youngeun Kweon, In Young Yoo, Eun-Jee Oh, Yeon-Joon Park, Yonggoo Kim, Hoon Seok Kim, Dong Wook Jekarl

**Affiliations:** 1Department of Laboratory Medicine, Seoul St. Mary’s Hospital, College of Medicine, The Catholic University of Korea, 222, Banpo-daero, Seocho-gu, Seoul 06591, Republic of Korea; jaekwon@catholic.ac.kr (J.K.K.);; 2Research and Development Institute for In Vitro Diagnostic Medical Devices, College of Medicine, The Catholic University of Korea, Seoul 06591, Republic of Korea

**Keywords:** platelet concentrates, apheresis platelet, blood product contamination, bacteria, RNA-sequencing, differential gene expression

## Abstract

Microbial contamination of platelet concentrates (PC) remains a persistent challenge in transfusion medicine, necessitating robust preventive measures prior to product release. Differentially expressed gene (DEG) analysis of microbe inoculated PC offers a promising approach to identifying potential biomarkers for contamination detection. Within PC, each *S. aureus* (ATCC 29213) and *S. epidermidis* (ATCC 12228) was inoculated in a 10^3^ CFU/mL concentration. Total RNA was extracted from the samples at predetermined time points (0-, 1-, 3-, and 6-hours post-inoculation), followed by high-throughput RNA sequencing. DEG, gene enrichment, and pathway analysis were conducted. Diagnostic potential was evaluated through the calculation of area under the curve (AUC) values and the assessment of additional performance metrics. DEG identified 5884 and 974 DEGs in *S. aureus* and *S. epidermidis* samples, respectively. Pathway analysis revealed distinct biological responses: *S. aureus*-inoculated samples showed prominent enrichment in ribosomal and spliceosome pathways, while *S. epidermidis*-inoculated samples demonstrated significant activation of mitogen-activated protein kinase (MAPK) signaling pathways and natural killer (NK) cell-mediated cytotoxicity pathways. ROC analysis of the commonly differentially expressed genes in both *S. aureus* and *S. epidermidis*-inoculated samples demonstrated significant diagnostic potential. The genes *H19*, *CAVIN1*, *A2M*, and *EPAS1* exhibited statistically significant adjusted *p*-values and AUC values exceeding 0.8, with the exception of the *H19* gene in *S. epidermidis*, suggesting their utility as potential biomarkers for staphylococcal contamination detection. Interaction between PC and microbial contaminants resulted in DEG and genes could be analyzed for microbial contamination of PC. However, to establish the robustness and broader applicability of these findings, further studies encompassing a more diverse range of microbial species are necessary.

## 1. Introduction

Platelet concentrates (PC) are widely utilized in transfusion medicine, particularly for patients with diverse clinical conditions. These include individuals with hematologic malignancies, oncology patients undergoing chemotherapy, those with hepatic disorders, and critically ill patients in intensive care units [1,2]. PC are maintained under specific storage conditions, namely at 20–24 °C with continuous agitation, to preserve cellular function, sustain metabolic processes, and ensure adequate oxygenation [3,4]. However, these room temperature storage conditions and relatively high oxygen provision particularly render PC susceptible to microbial proliferation, especially skin normal flora and environmental sources of bacteria [5]. Epidemiological data indicate that the incidence of microbial contamination in apheresis platelet units ranges from 1:1000 to 1:2500. Moreover, the frequency of fatal outcomes associated with contaminated platelet transfusions has been reported at approximately 1:108,000 units [6,7,8,9,10].

Donor skin microflora has been identified as the predominant source of microbial contamination in platelet components [11]. A comprehensive study analyzing primary cultures from 960,470 apheresis units revealed that 63.5% of bacterial contaminants were species associated with donor skin microbiota, while 24.5% originated from mucous membrane microbiota, and 12.0% from other sources [12]. The principal bacterial species derived from donor skin include *Staphylococcus aureus*, *Staphylococcus epidermidis*, various coagulase-negative staphylococci, and *Corynebacterium* species [13].

To mitigate the risk of microbial contamination, a multi-faceted approach has been implemented, encompassing rigorous donor selection protocols, aseptic skin preparation techniques, utilization of blood collection containers with integrated diversion pouches, implementation of pathogen reduction systems, and screening for potential microbial contamination using advanced assay methodologies [11,14,15,16,17,18].

Microbial contamination in platelet components can be detected through various methodologies, each with distinct characteristics. Culture-based methods, while highly sensitive, require a prolonged cultivation period to achieve their low detection limit (10^1^ to 10^2^ CFU/mL) [19]. Lateral flow assays offer rapid results by detecting microbial cell wall components via antigen-antibody reactions but necessitate a higher microbial load for detection compared to culture methods (analytical sensitivity 10^3^ to 10^5^ CFU/mL) [18]. Nucleic acid amplification tests (NAATs), targeting bacterial 16S or 23S ribosomal genes, provide a balance between test turnaround time and sensitivity, demonstrating faster turnaround times than culture-based techniques and lower detection limits than lateral flow assays (limit of detection of 13 ± 4.94 CFU/mL). However, NAATs require stringent method standardization and quality control measures to ensure reliability [18].

Next-generation RNA sequencing (RNA-seq) technology has significantly advanced our understanding of platelet biology by providing comprehensive transcriptomic profiles in various contexts. These include elucidating normal platelet gene expression patterns, characterizing platelet mRNA expression in sepsis patients, and monitoring transcriptional changes in platelet products during storage [20,21,22]. Despite these advances, the gene expression landscape of bacterially contaminated platelets remains largely unexplored. Explanation of the transcriptional changes associated with bacterial presence in platelet products is crucial for identifying potential biomarkers and understanding the molecular mechanisms underlying platelet-pathogen interactions [23,24,25,26,27].

In this study, we postulated that microbial contamination in platelet components (PC) would induce detectable alterations in RNA expression profiles. We hypothesized both white blood cells and platelets would engage in interactions with the introduced microbial agents, resulting in distinct transcriptional changes. To test this hypothesis, we employed RNA sequencing (RNA-seq) technology to analyze the transcriptome of PC at multiple time points following bacterial inoculation. Specifically, we examined the differential gene expression patterns at 0-, 1-, 3-, and 6-hours post-inoculation with *S. aureus* and *S. epidermidis*. This temporal analysis aimed to explicate the dynamic transcriptional changes occurring in response to bacterial contamination, potentially revealing early molecular indicators of microbial presence in PC.

## 2. Results

### 2.1. Quality Control Metrics

The quality control data from blood service including volume, pH, and platelet count of PC from 400 mL whole blood was as follows: 49 mL, pH 7.05, 6.6 × 10^10^/unit. Following bacterial inoculation, the final concentration was determined to be 2.03 CFU/mL.

The average number of total reads per sample was 114,319,791.3 for *S. aureus*, 122,060,437.4 for *S. epidermidis*, and 123,675,789.1 for the control group. For the *S. aureus* inoculation group, the mean mapped read count was 109,850,678, with a mapping percentage of 97.98%. The *S. epidermidis* inoculation samples demonstrated a mean mapped read count of 116,090,627, corresponding to a mapping percentage of 97.35%. The control group exhibited identical characteristics to the *S. aureus* group, with a mean mapped read count of 109,850,678 and a mapping percentage of 97.98%. The percentage of bases with a Phred quality score above 20 exceeded 97% for all samples. We use samples with a RIN greater than 7 [28,29].

### 2.2. Differentially Expressed Gene (DEG) of Samples After Inoculation of S. aureus

A total of 31,476 genes that undergo prefiltering were analyzed for differentially expressed gene analysis. The differential gene expression analysis revealed the following counts of up-regulated and down-regulated genes at 1-, 3-, and 6-hours post-treatment: 1 h (1528 up, 2071 down), 3 h (1305 up, 2162 down), and 6 h (867 up, 2022 down). Temporal analysis of differential gene expression across all time points identified a total of 5884 genes (3009 up, 2875 down) exhibiting significant changes when samples were stratified according to elapsed time. The DEGs between 0 and 1 h included up-regulated genes such as *EPAS1*, *CDKN1C*, *ZNF761*, *TYMP*, *CAVIN1*, and *NADK*, while down-regulated genes comprised *CCNYL1*, *EEF1AKMT1*, *DNAI4*, *HECTD2*, *RPPH1*, and *SPIDR*. Comparative analysis of gene expression between 0- and 3-hours post-treatment identified several DEGs. Among the up-regulated genes were *TYMP*, *H19*, *CAVIN1*, *URB2*, *IGF2*, and *A2M*, while *CCNYL1*, *RPS29*, *EXOC6B*, *EEF1AKMT1*, *SNORD133*, and *SPIDR* were found to be down-regulated. Further examination of DEGs between 0 and 6 h revealed up-regulation of *CAVIN1*, *IGF2*, *A2M*, *ARHGAP29*, *EPAS1*, and *H19*, whereas *EEF1AKMT1*, *CCNYL1*, *HECTD2*, *EXOC6B*, *ADAL*, and *AGAP4* exhibited down-regulation. The temporal expression patterns of differentially regulated genes are summarized (Table 1).

Principal component analysis (PCA) was performed to visualize the overall gene expression patterns throughout the observed time period (Figure 1A). To further elucidate the magnitude and statistical significance of gene expression changes, volcano plots were constructed using log2 fold change values plotted against −log10 *p*-values over the entire time course (Figure 1B). The supplementary data presents PCA plots and volcano plots for each time point (Appendix A). To delineate the expression patterns of DEGs, a hierarchical cluster analysis was conducted. The resulting patterns were visualized through a heatmap (Appendix A).

### 2.3. Comprehensive Gene Enrichment for Pathway Analysis of Samples After Inoculation of S. aureus

Gene set enrichment for pathway analysis was conducted on samples inoculated with *S. aureus* using the pathfindR package in R. The analysis utilized gene sets corresponding to the Homo sapiens (hsa) KEGG pathways. Results revealed significant enrichment in 225 pathways, notably ribosome, spliceosome, and nucleocytoplasmic transport pathways, all of which demonstrated high −log10 *p*-values, indicating strong statistical significance (Figure 1C). The term-gene graph of inoculated platelet concentrates shows both ribosome and spliceosome pathways had no overlapping genes (Appendix A). Further pathway analysis was conducted using the GeneTonic R package v.3.0.0 with Gene Ontology (GO) pathways. The resulting enrichment map revealed significant enrichment in several key biological processes, notably chromatin remodeling, cell division, and regulation of transcription elongation by RNA polymerase III (Appendix A). The gene set volcano plot corroborated the findings from the previous enrichment map, reaffirming the significance of chromatin remodeling and cell division processes. Additionally, this analysis revealed a novel pathway of interest: the proteasome-mediated ubiquitin-dependent protein catabolic process (Appendix A).

### 2.4. Differentially Expressed Gene (DEG) of Samples After Inoculation of S. epidermidis

A comprehensive analysis of differential gene expression was conducted on a total of 23,794 genes that passed the initial quality filtering criteria. Following the methodology applied to *S. epidermidis* inoculation samples, we identified DEGs at 1-, 3-, and 6-hours post-treatment, using stringent criteria, same as the analysis of samples with *S. aureus* inoculation. Differential gene expression analysis, performed across all time points, revealed 852 up-regulated and 122 down-regulated genes when samples were stratified by elapsed time. The temporal analysis revealed the following DEG counts: 1 h (10 up-regulated, 7 down-regulated), 3 h (2 up-regulated, 3 down-regulated), and 6 h (5 up-regulated, 3 down-regulated). At the 1 h time point, notable up-regulated genes included *SERPINE1*, *BPI*, *IL1R1*, *PTGS2*, *RPL10P9*, and *DNAH17*, while *HNRNPUL2-BSCL2*, *LOC105370464*, *BIVM-ERCC5*, *RPL23AP7*, *RPL23AP79*, and *LINC00964* were significantly down-regulated. The 3-hours post-treatment analysis identified *SLC34A3* and *RPL10P9* as up-regulated, whereas *TMX2-CTNND1*, *TRIM6-TRIM34*, and *LOC101927613* showed down-regulation. At 6 h post-treatment, *LMO3*, *RPL10P9*, *NXF3*, *PTGS2*, and *MXRA7* exhibited up-regulation, while *PRSS1*, *RPL23AP7*, and *RPL23AP79* were down-regulated. The significant transcriptional changes observed in the *S. epidermidis* inoculation samples throughout the period are summarized (Table 2).

PCA plots (Figure 2A) and volcano plots (Figure 2B) were generated across all time points. Additionally, the Appendix A include PCA plots and volcano plots corresponding to each time point (Appendix A).

### 2.5. Comprehensive Gene Set Enrichment for Pathway Analysis of Samples After Inoculation of S. epidermidis

Same as *S. aureus* analysis, gene set enrichment for pathway analysis was performed on *S. epidermidis*-inoculated samples utilizing the pathfindR package in R, employing the same Homo sapiens (hsa) KEGG pathway database. This comprehensive analysis identified 177 significantly enriched pathways. Among these, several key cellular signaling and immune response pathways demonstrated particularly strong enrichment, including the MAPK signaling pathway, proteoglycans in cancer, NK cell-mediated cytotoxicity, and focal adhesion pathways (Figure 2C). Analysis of the term-gene network in *S. epidermidis*-inoculated samples revealed substantial gene overlap among four distinct pathways, with key shared components including *PAK1*, *RAF1*, *GRB2*, and *ARAF* (Appendix A). Subsequent pathway analysis, conducted using the GeneTonic R package in conjunction with GO pathways, generated an enrichment map highlighting several significant biological processes: positive regulation of interleukin-6 (IL-6) production, negative regulation of cell population proliferation, symbiont entry into the host cell, and inflammatory response (Appendix A). The gene set volcano plot analysis further validated these findings, particularly emphasizing the significance of positive regulation of IL-6 production and negative regulation of cell population proliferation (Appendix A).

### 2.6. DEG Analysis and ROC Curve Assessment of Commonly Expressed Genes in S. aureus and S. epidermidis Inoculated Samples

Differential expression analysis revealed a total of 403 genes exhibiting significant expression alterations in platelet concentrate samples inoculated with both *S. aureus* and *S. epidermidis*. Furthermore, we present a comprehensive overview of the top 10 differentially expressed genes that demonstrated consistent expression patterns across both bacterial contamination conditions. Each of these identified genes met the statistical significance threshold criterion (adjusted *p*-value < 0.05) in both the *S. aureus* and *S. epidermidis* inoculated samples (Table 3). Subsequently, we selected the four genes (*H19*, *CAVIN1*, *A2M*, and *EPAS1*) exhibiting the most statistically significant *p*-values in both *S. aureus* and *S. epidermidis* inoculated samples for further evaluation through ROC curve analysis.

Receiver Operating Characteristic (ROC) curve analysis was performed on the four candidate genes that we manually selected based on their consistent differential expression patterns in both *S. aureus* and *S. epidermidis* contaminated samples. The analysis demonstrated that these four candidate genes possess substantial diagnostic potential, with area under the curve values exceeding 0.8 (AUC > 0.8) for all selected genes except H19, which nevertheless approached this threshold (Figure 3). Each of these manually curated candidate genes exhibited distinct and statistically robust optimal cutoff values, which could be instrumental in establishing quantitative parameters for the development of highly sensitive and specific diagnostic assays.

### 2.7. Multiseries Time-Course Analysis of Gene Expression Patterns and ROC Curve Generation for Curated Genes in S. aureus and S. epidermidis Inoculated Specimens

Utilizing the maSigPro R package v1.78.0, we systematically elucidated and visualized the time-specific gene expression profiles in samples inoculated with *S. aureus*. Our analysis identified a total of 2439 genes that met our significance criteria, with *p*-values less than 0.05 and R-squared values exceeding 0.5, indicating robust temporal expression patterns. From this subset, we further refined our focus to select four candidate genes (*YY1AP1*, *PDSS1*, *CDC25C*, *CCZ1B*) that demonstrated particularly high statistical significance and strong model fit (Table 4). Notably, the expression patterns of these genes exhibited a clear differential between the S. aureus inoculated group and the control group, as evidenced by the distinct profiles in the generated plots (Figure 4).

Following the meticulous curation of candidate genes, we generated ROC curves to evaluate their discriminative performance (Figure 5). The analysis demonstrated that each candidate gene achieved an AUC of 1, indicating a perfect separation between the *S. aureus* inoculated group and the control group. Logistic regression models for all candidate genes could not be established due to the perfect separation in the data. These findings suggest that these genes could serve as highly sensitive and specific biomarkers for detecting *S. aureus* contamination in platelet concentrates.

Employing identical methodological approaches and selection criteria as those utilized for *S. aureus*-inoculated specimens, we identified seven genes exhibiting statistically significant differential expression. From this subset, we subsequently selected four genes (*FARS2*, *LOC105370462*, *LOC105377460*, and *ZCCHC7*) that demonstrated consistently elevated expression levels across temporal expression profiles (Table 5, Figure 6).

Subsequent to the rigorous selection of candidate genes, we conducted ROC curve analyses to assess their discriminatory efficacy (Figure 7). Unlike the *S. aureus* experimental group, perfect separation between control and *S. epidermidis*-inoculated specimens was not achieved, resulting in comparatively reduced AUC values. Nevertheless, with the exception of the *FARS2* gene, all candidate genes demonstrated robust AUC values exceeding 0.9. These findings suggest that the curated genes may serve as appropriate diagnostic biomarkers for bacterial contamination in platelet concentrates, particularly for the detection of *S. epidermidis* contamination.

## 3. Discussion

The predominant microorganisms associated with PC contamination are *Staphylococcus* and *Streptococcus* species, which are common constituents of the normal skin flora [7,8]. These microorganisms not only serve as primary contaminants of platelet products but have also been increasingly implicated in transfusion-associated morbidity and mortality. While pathogen inactivation technologies have been implemented as preventive measures against platelet contamination, they have not proven uniformly effective. Notable cases have documented fatal outcomes in PC recipients due to contamination with *Acinetobacter* and *Staphylococcus* species, even in pathogen-reduced platelet products [9,10].

Early detection of microbial contamination could facilitate the identification of compromised PC prior to their distribution from blood services to hospitals or from blood banks to patients. We hypothesized that PC contamination might induce differential gene expression in residual white blood cells or platelets within the concentrate. Indeed, our analysis revealed distinct transcriptional changes at 1, 3, and 6 hours post-*S. aureus* and *S. epidermidis* inoculation compared to baseline (0 h). The inoculation dose was standardized at 10^3^ CFU/mL, corresponding to the detection threshold of conventional lateral flow assays [18].

The observed transcriptional modifications likely reflect molecular interactions between contaminating microorganisms and cellular components within the PC. Given that white blood cells contain approximately 10^3^-fold higher mRNA content, the RNA-seq analysis presumably captures gene expression from both white blood cells and platelets. The PTPRC gene encoding CD45 serves as a white blood cell marker, while PF4 is specifically expressed in platelets, allowing assessment of white blood cell presence [27]. Previous research by Middleton et al. demonstrated that in platelet concentrates sequencing, FPKM values of the PTPRC gene were approximately 5000-fold lower than PF4, indicating minimal white blood cell contamination. In our analysis, the mean FPKM values across all samples were 2539 for PF4 and 7.878 for PTPRC, indicating that the FPKM values of the PTPRC gene were approximately 300-fold lower than that of PF4. This finding suggests that white blood cell contamination was not entirely eliminated in our experimental samples.

In the current investigation, we conducted an ROC curve analysis to assess the diagnostic utility of hub genes identified from the four most significantly enriched pathways in *S. epidermidis* inoculated samples. We initially selected three genes (*PAK1*, *RAF1*, and *GRB2*) based on their involvement in critical molecular signaling networks, including the MAPK signaling pathway, proteoglycans in cancer, focal adhesion, and natural killer cell-mediated cytotoxicity. However, all three genes exhibited statistically non-significant *p*-values (≥0.05) accompanied by suboptimal area under the curve (AUC) values. Based on these findings, we determined that a hub gene-centric approach was not appropriate for achieving the objectives of our study. Consequently, we implemented an alternative methodology, utilizing statistical significance (*p*-values) and discriminatory capacity (AUC values) as primary selection criteria for candidate gene identification. These statistical metrics were generated through differential expression analysis using the DESeq2 algorithm.

Comparative analysis identified 403 genes with significant differential expression (*p*-value < 0.05) in both *S. aureus* and *S. epidermidis* samples, including *H19*, *CAVIN1*, *A2M*, and *EPAS1*. The observed transcriptional changes align with expected biological responses, as some of these genes are associated with inflammatory and immune responses. For instance, the H19 gene encodes a long non-coding RNA that plays a significant regulatory role in inflammation, aging, and disease progression [30]. Studies have demonstrated that H19 is involved in the modulation of inflammatory pathways, particularly in arthritic conditions. Furthermore, H19 exhibits increased expression in various malignancies, functioning as a competitive endogenous RNA that sequesters microRNAs, thereby altering the expression profiles of downstream target genes. Alpha-2-macroglobulin (A2M) functions as a broad-spectrum protease inhibitor that employs a distinctive “bait-and-trap” mechanism [31]. A2M demonstrates significant immunomodulatory properties, including enhancement of neutrophil functionality through promotion of adhesion to endothelial cells, facilitation of migration, and augmentation of pathogen phagocytosis and elimination. Additionally, A2M stimulates bacterial phagocytosis, ROS production, and antigen presentation via major histocompatibility complex class I and II (MHC-I and MHC-II) pathways. Furthermore, A2M plays a crucial role in the regulation of fibrinolysis and coagulation processes through its inhibition of proteinases, a phenomenon that has been documented in patients with sepsis [32]. The EPAS1 gene, also known as HIF-2α, serves a pivotal role in the regulation of physiological responses to hypoxic conditions. These hypoxic states may be precipitated by high-altitude exposure or pathological conditions such as sepsis, which characteristically manifests with tissue hypoxia and inflammatory processes [33].

We compared our findings with previously published RNA sequencing data from platelets of sepsis patients [26]. Temporal analysis of gene expressions after bacterial inoculation revealed significant upregulation of *H19*, *A2M*, and *EPAS1* genes in both *S. aureus* and *S. epidermidis* inoculated platelet concentrates. While this gene also showed elevated expression in sepsis patients, the increases did not reach statistical significance. Conversely, *LOC105376872* and *SNORA53* genes demonstrated significant downregulation in both samples, but neither gene was identified among the downregulated genes in sepsis patients based on transcriptomic analysis. The observed differences in expression patterns may be attributable to the distinct sample types used: transfusion-ready platelet-rich plasma in our study versus platelet mRNA in sampled monovettes in the previous investigation. Previous studies utilizing a sepsis mouse model demonstrated the upregulation of the *ITGA2B* gene, which correlated with increased mortality rates [27]. In contrast, our RNA sequencing analysis of microorganism-inoculated platelet concentrates failed to detect significant alterations in *ITGA2B* gene expression levels.

Ribosomes play a fundamental role in protein synthesis, utilizing mRNA as a template and amino acids as substrates [34]. Ribosomal dysfunction has been implicated in various pathological conditions, including altered bacterial resistance, SARS-CoV-2 infection, and oncogenesis [35,36]. Our analysis revealed statistically significant enrichment of the ribosomal pathway in *S. aureus*-inoculated samples, suggesting its involvement in the immunoregulatory response to bacterial contamination. The MAPK family, comprising serine/threonine protein kinases, regulates essential biological processes and cellular responses to external stressors [37]. Enhanced MAPK activity, particularly p38 MAPK, regulates inflammatory mediator synthesis at both transcriptional and translational levels. Notably, our analysis of *S. epidermidis*-inoculated samples demonstrated significant enrichment of the MAPK signaling pathway. NK cells, classified as innate lymphoid cells, possess the capacity to eliminate virally infected and tumor cells [38]. These cells employ an HLA-unrestricted recognition mechanism, utilizing diverse arrays of activating and inhibitory receptors to discriminate between healthy and infected cells. As group 1 innate lymphoid cells, NK cells respond to both intracellular and extracellular bacteria and demonstrate lytic activity against various bacterial species [39,40,41]. Significantly, pathway analysis of *S. epidermidis*-inoculated samples revealed prominent enrichment of the NK cell-mediated cytotoxicity pathway.

ROC analysis identified four candidate genes (*H19*, *CAVIN1*, *A2M*, and *EPAS1*) that were selected based on DESeq2 results with significant *p*-values and notable AUC values for distinguishing between normal and inoculated platelet concentrates (Table 3). These genes demonstrated high area under the curve values in both *S. aureus* and *S. epidermidis* inoculated samples (AUC > 0.8), with the exception of the H19 gene, which nonetheless approached this threshold (AUC = 0.8). These findings suggest their potential utility as early diagnostic biomarkers for the detection of *S. aureus* and *S. epidermidis* contamination in platelet concentrates.

Despite our comprehensive approach, we were unable to identify common genes between *S. aureus* and *S. epidermidis* samples in the time-domain analysis utilizing the maSigPro algorithm. This limitation may be attributed to several factors, including the stringent *p*-value and R-squared value criteria employed in our analytical framework, potential procedural inconsistencies, or insufficient temporal correlation between gene expression levels in the *S. epidermidis* inoculated samples. While we acknowledge that the rigorous statistical thresholds implemented were instrumental in maintaining analytical integrity, we also recognize that relaxing these parameters could potentially compromise the validity and reliability of our findings. Accordingly, we conducted a targeted selection process by independently identifying four distinct candidate genes from *S. aureus* inoculated specimens and four separate candidate genes from *S. epidermidis* inoculated specimens. This gene selection methodology was implemented separately from, and not influenced by, the candidate genes previously identified through the DESeq2 differential expression analysis. Therefore, we regarded the DESeq2 as the main analytical approach to identify commonly differentially expressed genes between *S. aureus* and *S. epidermidis* inoculated samples, circumventing the limitations encountered with the maSigPro time-series analysis.

Several limitations should be acknowledged in the present study. First, our investigation was restricted to *S. aureus* and *S. epidermidis* contamination, while other significant PC contaminants, including *Streptococcus* species and gram-negative bacteria, were not examined. Additionally, a methodological limitation emerges from our utilization of ATCC strains of *S. aureus* and *S. epidermidis* rather than bacterial isolates derived from human skin flora. This methodological discrepancy may potentially induce differential RNA expression patterns during contamination and proliferation phases, thereby potentially affecting the accuracy and clinical translatability of our diagnostic assay results. Also, the study was limited by our inability to quantify bacterial loads at sequential time points throughout the experimental period. This limitation is attributable to the technical constraints of our methodology, specifically establishing the limit of detection. Furthermore, bacterial growth kinetics and potential co-contamination scenarios were not evaluated. Future studies should encompass these other clinically relevant microorganisms and investigate various inoculation concentrations, particularly around 1 CFU/mL, which represents the detection threshold for conventional culture methods. Furthermore, the platelet concentrates utilized in this study were obtained from donors with elevated AST values (2–3 times above the normal range). The possibility of underlying liver disease, including hepatitis B virus infection, could not be definitively excluded. These varying hepatic conditions might have influenced the observed gene expression patterns, potentially confounding our results.

## 4. Materials and Methods

### 4.1. Sample Preparation

This study was approved by the institutional review board of Seoul St. Mary’s Hospital and Hanmaeum blood service (Gunpo, Gyeonggi-do, Korea). PC prepared by the platelet-rich plasma method, which exhibited aspartate aminotransferase (AST) levels exceeding 100 IU/mL and thus did not meet the quality criteria for transfusion, were utilized in this study. PC products were transported to the blood bank of Seoul St. Mary’s Hospital. Upon arrival, each PC unit was inoculated with 100 μL of *S. aureus* (ATCC 29213) and *S. epidermidis* (ATCC 12228) suspension at a concentration of 10^3^ CFU/mL. This concentration was selected because in this concentration, microbes growth are expected to be detected after 12 h of culture. The inoculated PC were stored in a platelet agitator and incubator under standard conditions. RNA extraction was performed using 6 mL of platelet-rich plasma collected in EDTA tubes (BD Biosciences, San Jose, CA, USA) at the following time points: immediately before inoculation (0 h), and at 1-, 3-, and 6-hours post-inoculation. A total of 75 samples were prepared, consisting of 27 samples in the *S. aureus* group, 24 samples in the *S. epidermidis* group, and 24 samples in the control group.

### 4.2. RNA Preparation, Library Construction, and Sequencing

RNA from platelet-rich plasma was extracted based on previous methods using an RNA extraction kit (Roche Diagnostics, Basel, Switzerland) [23,24]. cDNA synthesis, library preparation, and RNA-seq were performed by Macrogen (Macrogen Inc., Seoul, Republic of Korea). In brief, total RNA was isolated from the samples and subjected to DNase treatment to eliminate DNA contamination, followed by purification using TruSeq Stranded Total RNA LT Sample Prep Kit (Illumina Inc., San Diego, CA, USA). For short-read sequencing, the purified RNA was fragmented, and cDNA synthesis was performed. Subsequently, adaptor ligation, PCR amplification, and sequencing were carried out using Illumina NovaSeq 6000 (Illumina Inc., San Diego, CA, USA). RNA quality was measured by RNA integrity number (RIN) [28].

### 4.3. Data Quality Control and Read Mapping

The quality of raw transcriptome data was evaluated using multiple metrics, including total read bases, total reads, and GC content. Additionally, sequencing accuracy was assessed by calculating Q20 and Q30 scores based on the Phred quality score using fastp (v0.23.0). The raw FASTQ files were subsequently processed using fastp for adapter trimming and quality filtering. Adapter sequences were removed, and low-quality bases (Phred score < 3) were discarded. Reads shorter than 36 bp were excluded to ensure reliable downstream analyses. Following preprocessing, the filtered reads were aligned to the human reference genome (GRCh38) using HISAT2 (v2.2.1). The resulting SAM files were converted into BAM format using samtools (v1.19). Transcript quantification was performed using StringTie (v2.2.1), which estimated gene expression levels based on the reference gene model. Gene expression levels were quantified as read counts and fragments per kilobase of transcript per million mapped reads (FPKM). To reduce potential bias from lowly expressed genes in downstream statistical analyses, genes with read counts below 10 were excluded using a threshold-based filtering approach [42]. All analyses were conducted in an Ubuntu (v22.04.4) computing environment.

### 4.4. Differential Gene Expression Analysis

Differential expression analysis was conducted utilizing DESeq2 R package version 1.44.0 with default parameters [43], a package that offers statistical approaches for identifying differentially expressed genes in digital transcriptome data through a model grounded in negative binomial distribution. Genes characterized by |log_2_(fold change)| > 1 and adjusted *p*-value < 0.05 were considered as differentially expressed genes. Genes were classified as up-regulated when exhibiting a log_2_(fold change) > 1 and adjusted *p*-value < 0.05, while genes with log_2_(fold change) < −1 and adjusted *p*-value < 0.05 were designated as down-regulated. For data visualization, we employed multiple techniques. Principal Component Analysis (PCA) plots were created using the plotPCA function from the DESeq2 package in R. For enrichment analysis utilizing active subnetworks and subsequent clustering of enrichment terms, we employed the pathfindR R package v2.4.2 [44]. Extracted pathways were annotated with the Kyoto Encyclopedia of Genes and Genomes (KEGG).

To comprehensively analyze our specimens while integrating time-course data, we employed the maSigPro R package, which utilizes generalized linear models to systematically detect temporal patterns in gene expression. Following model fitting, genes were meticulously ranked based on their *p*-values and R-squared metrics, thereby providing a dual assessment of both statistical significance and the goodness-of-fit of the temporal models. Moreover, candidate genes were selected not only on the basis of these quantitative criteria but also through detailed graphical analysis, which confirmed that these genes consistently exhibited elevated expression levels across all time points. This comprehensive strategy ensured that the final selection of candidate genes was both statistically robust and biologically meaningful.

Following the careful curation of candidate genes, we generated ROC curves to assess their discriminative performance. Subsequently, we calculated the AUC values and determined the confidence intervals for these AUCs and optimal cutoff thresholds using the ROCit R package v2.1.2. This analysis was exclusively performed on genes that exhibited consistent expression profiles throughout the entire time course, thereby reinforcing the robustness and reliability of our biomarker selection process.

## 5. Conclusions

This study investigated differential gene expression patterns between bacterially contaminated and control platelet concentrate samples. Our analysis identified 5884 differentially expressed genes in *S. aureus*-inoculated samples and 974 in *S. epidermidis*-inoculated samples. Subsequent gene enrichment analysis revealed significant enrichment of multiple pathways associated with infection and inflammatory responses. ROC analysis of both bacterial groups demonstrated consistent upregulation of *H19*, *CAVIN1*, and *EPAS1* genes, suggesting their potential utility as biomarkers for predicting platelet concentrate contamination. Additionally, the incorporation of time-series analysis highlighted the significance of additional genes, broadening our understanding of the transcriptional response to bacterial contamination.

## Figures and Tables

**Figure 1 ijms-26-03009-f001:**
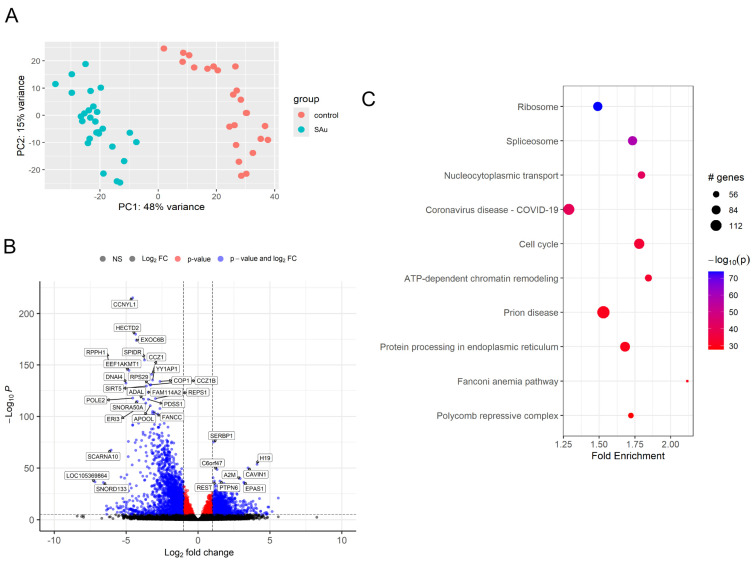
(**A**) Principal component analysis (PCA) plots of gene expression profiles of platelet concentrate following *S. aureus* inoculation over the entire time course. (**B**) Volcano plots of DEGs in the *S. aureus* inoculated platelet concentrate across all time points. (**C**) Pathway analysis of samples inoculated with *S. aureus*. The results revealed highly significant enrichment in several key cellular processes, including ribosome, spliceosome, and nucleocytoplasmic transport pathways.

**Figure 2 ijms-26-03009-f002:**
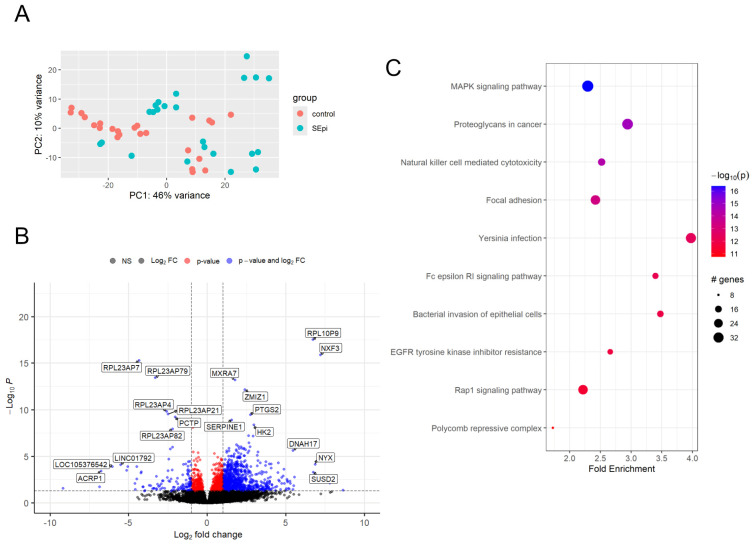
(**A**) PCA plots of gene expression profiles of platelet concentrate after *S. epidermidis* inoculation. (**B**) Volcano plots of DEGs in the *S. epidermidis* inoculated platelet concentrate. (**C**) Pathway analysis of samples inoculated with *S. epidermidis*. The results revealed highly significant enrichment in several key cellular processes, including MAPK signaling pathway, proteoglycans in cancer, NK cell-mediated cytotoxicity, and focal adhesion.

**Figure 3 ijms-26-03009-f003:**
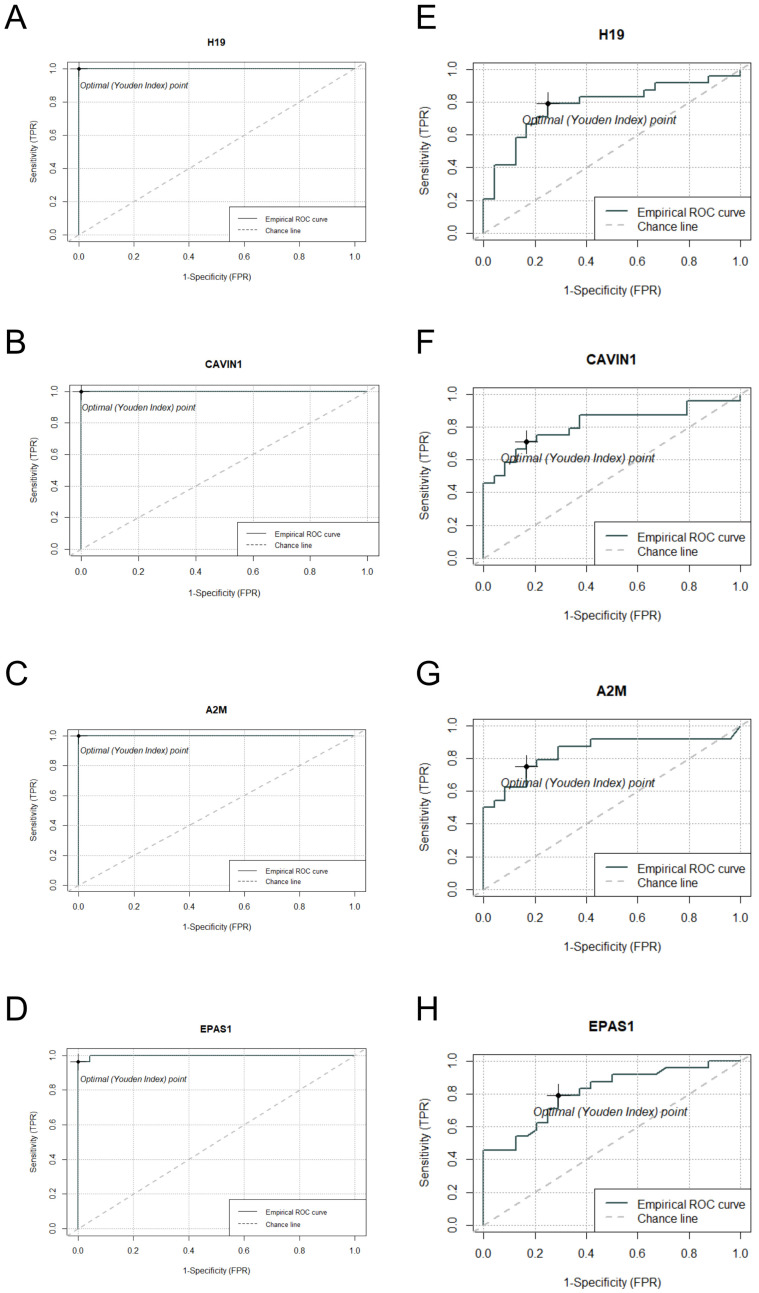
ROC curve analysis of four manually curated candidate genes. Panels (**A**–**D**) represent the ROC curves for the selected genes in *S. aureus* inoculated samples, while panels (**E**–**H**) illustrate the ROC curves for the same genes in *S. epidermidis* inoculated samples. The analysis demonstrates the diagnostic potential of these genes for detecting staphylococcal contamination in platelet concentrates.

**Figure 4 ijms-26-03009-f004:**
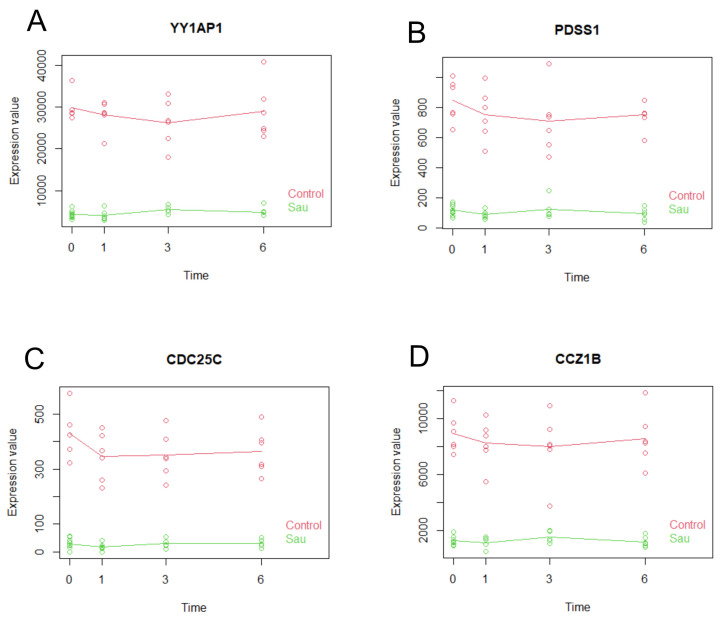
The gene expression plots for the four curated candidate genes consistently demonstrated marked differences between the *S. aureus* inoculated group and the control group across all assessed time points. (**A**–**D**) represent *YY1AP1*, *PDSS1*, *CDC25C*, and *CCZ1B* expression level, respectively. Each data point corresponds to a specific sample’s temporal position and measured expression level, while the linear element represents the collective expression trajectory of the group.

**Figure 5 ijms-26-03009-f005:**
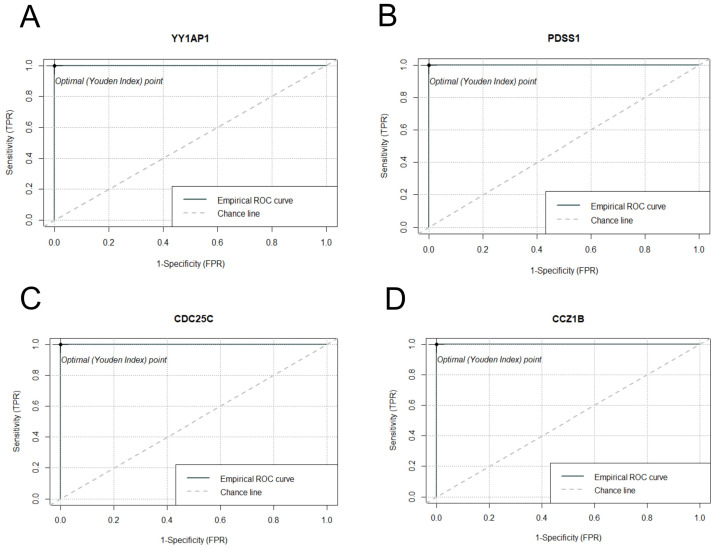
ROC curves of four curated genes in samples inoculated with *S. aureus*. (**A**–**D**) show ROC curves of *YY1AP1*, *PDSS1*, *CDC25C*, *CCZ1B* genes, respectively.

**Figure 6 ijms-26-03009-f006:**
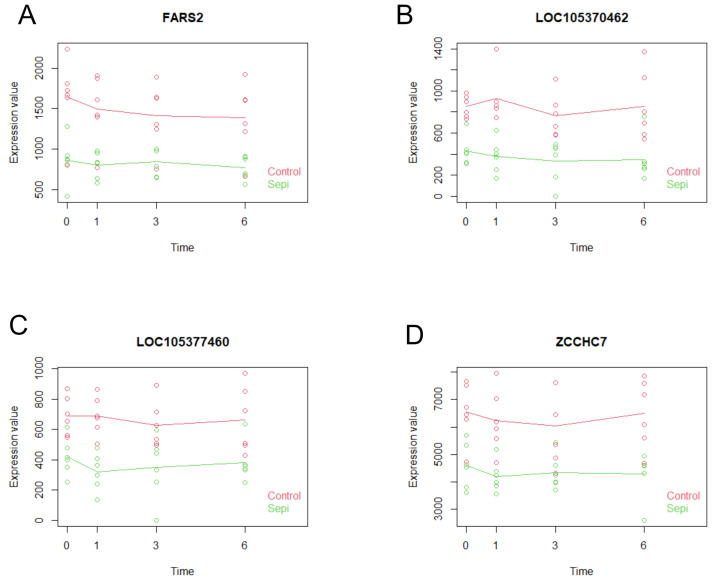
Expression profiles of the four selected candidate genes demonstrating consistent and pronounced differential expression between the *S. epidermidis*-inoculated group and the control group across all temporal measurement points. (**A**–**D**) represent *FARS2*, *LOC105370462*, *LOC105377460*, and *ZCCHC7* expression, respectively. Individual data points represent the temporal coordinates and corresponding expression values of specific samples, whereas the linear line illustrates the aggregate expression profile across the collective group.

**Figure 7 ijms-26-03009-f007:**
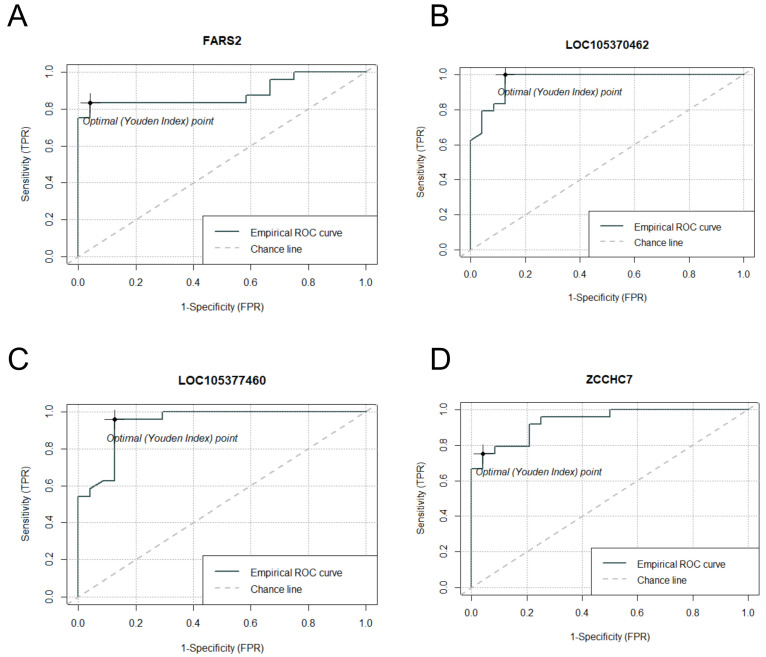
ROC curves for the four selected candidate genes in specimens inoculated with *S. epidermidis*. (**A**–**D**) depict the ROC curves for *FARS2*, *LOC105370462*, *LOC105377460*, and *ZCCHC7* genes, respectively.

**Table 1 ijms-26-03009-t001:** Up-regulated genes and down-regulated genes in *S. aureus* inoculated samples.

**Up-regulated genes at *S. aureus* inoculation**	
1 h	*EPAS1*, *CDKN1C*, *ZNF761*, *TYMP*, *CAVIN1*, *NADK*
3 h	*TYMP*, *H19*, *CAVIN1*, *URB2*, *IGF2*, *A2M*
6 h	*CAVIN1*, *IGF2*, *A2M*, *ARHGAP29*, *EPAS1*, *H19*
**Down-regulated genes at *S. aureus* inoculation**	
1 h	*CCNYL1*, *EEF1AKMT1*, *DNAI4*, *HECTD2*, *RPPH1*, *SPIDR*
3 h	*CCNYL1*, *RPS29*, *EXOC6B*, *EEF1AKMT1*, *SNORD133*, *SPIDR*
6 h	*EEF1AKMT1*, *CCNYL1*, *HECTD2*, *EXOC6B*, *ADAL*, *AGAP4*

**Table 2 ijms-26-03009-t002:** Up-regulated genes and down-regulated genes in *S. epidermidis* inoculated samples.

**Up-regulated genes at *S. epidermidis* inoculation**	
1 h	*SERPINE1*, *BPI*, *IL1R1*, *PTGS2*, *RPL10P9*, *DNAH17*
3 h	*SLC34A3*, *RPL10P9*
6 h	*LMO3*, *RPL10P9*, *NXF3*, *PTGS2*, *MXRA7*
**Down-regulated genes at *S. epidermidis* inoculation**	
1 h	*HNRNPUL2-BSCL2*, *LOC105370464*, *BIVM-ERCC5*, *RPL23AP7*, *RPL23AP79*, *LINC00964*
3 h	*TMX2-CTNND1*, *TRIM6-TRIM34*, *LOC101927613*
6 h	*PRSS1*, *RPL23AP7*, *RPL23AP79*

**Table 3 ijms-26-03009-t003:** Top 10 differentially expressed genes found commonly on two sample groups. Sau, *S. aureus*; Sepi, *S. epidermidis*; padj, adjusted *p*-value.

Gene Name	log2FC.Sau	padj.Sau	AUC	log2FC.Sepi	padj.Sepi	AUC
*H19*	4.104081	2.38 × 10^−54^	1	1.389704	0.004315	0.7795
*CAVIN1*	3.47354	6.56 × 10^−51^	1	1.477886	0.000469	0.8125
*A2M*	2.925464	1.82 × 10^−40^	1	1.178996	0.008684	0.8385
*EPAS1*	3.189499	4.96 × 10^−37^	0.9985	1.233558	0.000808	0.8090
*LOC105376872*	−4.28848	1.36 × 10^−33^	1	−1.04865	0.010769	0.8359
*SNORA53*	−4.63863	2.32 × 10^−31^	0.9598	1.510768	0.010982	0.6354
*PPM1F*	1.802254	1.14 × 10^−26^	0.9869	1.100995	0.008567	0.7118
*CDKN1C*	2.279073	3.42 × 10^−26^	0.9830	1.157439	0.001252	0.8056
*HK3*	2.435473	5.62 × 10^−26^	0.9923	1.328566	0.002777	0.7682
*RXRA*	2.338701	6.99 × 10^−26^	0.9799	1.5613	0.0002	0.7448

**Table 4 ijms-26-03009-t004:** Four candidate genes were identified using the maSigPro R package for subsequent downstream analyses. These genes were selected based on their robust statistical performance, as evidenced by high R-squared values and statistically significant *p*-values. Notably, ROC analysis demonstrated that the AUC for these genes was 1, indicating a perfect separation of read counts between the *S. aureus* inoculated group and the control group.

	*p*-Value	R-Squared	AUC	ciAUC	Cutoff
*YY1AP1*	1.45 × 10^−29^	0.9274	1	1–1	17,985
*PDSS1*	8.69 × 10^−26^	0.8965	1	1–1	470
*CDC25C*	1.12 × 10^−25^	0.8955	1	1–1	231
*CCZ1B*	1.29 × 10^−25^	0.8949	1	1–1	4065

**Table 5 ijms-26-03009-t005:** Four candidate genes selected from the subset of seven genes demonstrating statistical significance and consistently elevated expression levels across all temporal intervals. In contrast to the findings observed in *S. aureus* specimens, these candidates exhibited comparatively lower R-squared values and AUC values.

	*p*-Value	R-Squared	AUC	ciAUC	Cutoff
*FARS2*	4.67 × 10^−9^	0.5294	0.8854	0.7875–0.9833	1219
*LOC105370462*	9.83 × 10^−11^	0.6011	0.9696	0.9191–1	544
*LOC105377460*	1.01 × 10^−8^	0.5137	0.9418	0.8718–1	494
*ZCCHC7*	5.09 × 10^−9^	0.5277	0.9358	0.8622–1	5575

## Data Availability

Available data is contained within the article or Appendix A.

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
