# Peer review of "Alterations in RNA Expression Profile Following S. aureus and S. epidermidis Inoculation into Platelet Concentrates"

_ijms, 2025, doi:10.3390/ijms26073009_

Round 1
Reviewer 1 Report
Comments and Suggestions for Authors
This study aims to use RNA-seq to detect changes in human gene expression in platelet concentrates contaminated with Staphylococcus aureus and Staphylococcus epidermidis, with the goal of identifying a suitable gene marker for detecting microbial contamination in platelets. The research itself is intriguing, but I have a few concerns:
- The authors identified 14 and 18 candidate genes for S. aureus and S. epidermidis, respectively. How were these genes selected? The manuscript only mentions that the selection was based on ROC analysis and p-values. The authors should provide a more detailed explanation of the criteria used to prioritize these candidate genes.
- This study utilized standard strains (ATCC 29213 and ATCC 12228). Would the results be consistent if different clinical strains were used? It is recommended that skin-derived strains be included in the analysis to assess the robustness of the findings.
- The authors state, "each PC unit was inoculated with 100 μL of S. aureus (ATCC 29213) and S. epidermidis (ATCC 12228) suspension at a concentration of 10³ CFU/mL. This concentration was selected because, at this concentration, microbial growth is expected to be detectable after 12 hours of culture." What was the final bacterial concentration after inoculation? Measuring bacterial counts at 0, 1, 3, and 6 hours post-inoculation would help determine whether the detection limit of the proposed method is superior to existing approaches.
- The candidate gene selection process combined data from different time points. Could this lead to the inclusion of genes that only exhibit significant changes at specific time points? If so, this may limit the practical applicability of the proposed gene markers for contamination detection.
Author Response
Comments 1: The authors identified 14 and 18 candidate genes for S. aureus and S. epidermidis, respectively. How were these genes selected? The manuscript only mentions that the selection was based on ROC analysis and p-values. The authors should provide a more detailed explanation of the criteria used to prioritize these candidate genes.
Response 1: We are in agreement with this assertion. In the present study, we performed ROC curve analysis to evaluate hub genes from the top four pathways in S. epidermidis samples. Three genes (PAK1, RAF1, and GRB2) were initially selected based on their association with critical molecular pathways, including MAPK signaling pathway, proteoglycans in cancer, focal adhesion, and NK cell-mediated cytotoxicity. All three genes demonstrated p-values ≥0.05 with low AUC values. Consequently, we concluded that the hub gene-based approach was not suitable for the objectives of our investigation.
To effectively incorporate time-course data into our analysis, we employed the maSigPro R package to examine our dataset. The analysis yielded p-values and R-squared values, which served as quantitative parameters for the identification of novel candidate genes. The manuscript has been amended with modifications to lines 155-170 and Section 3.7.
In samples inoculated with S. aureus, a total of 2,439 genes demonstrated statistical significance with p-values less than 0.05 and R-squared values exceeding 0.5. Conversely, in S. epidermidis-inoculated samples, only 7 genes satisfied these same statistical criteria. For subsequent ROC analysis, we selected 4 candidate genes from the S. aureus dataset and also 4 candidate genes from the S. epidermidis dataset.
The selection criteria for candidate genes were established based on high R-squared values and consistent elevation in expression profiles throughout the time-series analysis in S. aureus and S. epidermidis samples.
The novel analytical framework we have developed supersedes our previous methodology for candidate gene identification, which relied solely on p-values and AUC metrics. This refined approach offers a more comprehensive and statistically robust method for prioritizing genes of interest in differential expression analyses.
Comments 2: This study utilized standard strains (ATCC 29213 and ATCC 12228). Would the results be consistent if different clinical strains were used? It is recommended that skin-derived strains be included in the analysis to assess the robustness of the findings.
Response 2: We appreciate your valuable suggestion and fully acknowledge its significance. However, obtaining S. aureus and S. epidermidis from human skin samples rather than utilizing ATCC reference strains presents several challenges. Specifically, the process requires collection of specimens from human subjects and subsequent bacterial culture, all of which involve considerable complexity and necessitate strict adherence to regulatory standards.
In the present study, we acknowledge a methodological limitation regarding the utilization of ATCC strains of S. aureus and S. epidermidis rather than bacterial isolates derived from human skin flora. Consequently, we have incorporated these considerations into the limitations section(line 571-579) at the discussion to acknowledge the potential constraints of our methodological approach.
Comments 3: The authors state, "each PC unit was inoculated with 100 μL of S. aureus (ATCC 29213) and S. epidermidis (ATCC 12228) suspension at a concentration of 10³ CFU/mL. This concentration was selected because, at this concentration, microbial growth is expected to be detectable after 12 hours of culture." What was the final bacterial concentration after inoculation? Measuring bacterial counts at 0, 1, 3, and 6 hours post-inoculation would help determine whether the detection limit of the proposed method is superior to existing approaches.
Response 3: The final bacterial concentration following inoculation was determined to be 2.03 CFU/mL. We added this information at Section 3.1, line 175-176. However, accurately measuring bacterial counts at 1, 3, and 6 hours post-inoculation presents significant challenges. This is primarily due to the labor-intensive nature of conventional culture-based methods, which require stringent experimental conditions, specialized equipment, and considerable time and effort. Given these constraints, obtaining precise bacterial counts at these specific time points remains technically demanding. The limitation regarding the absence of bacterial enumeration at multiple post-inoculation time intervals has been incorporated into the limitations section of the discussion.
Comments 4: The candidate gene selection process combined data from different time points. Could this lead to the inclusion of genes that only exhibit significant changes at specific time points? If so, this may limit the practical applicability of the proposed gene markers for contamination detection.
Response 4: As addressed in our response to Comment and Suggestion 1, we employed the maSigPro R package to comprehensively integrate and elucidate temporal alterations in gene expression profiles. Although our analysis identified numerous genes with statistically significant p-values and elevated R-squared values, we strategically selected candidate genes exhibiting consistently high expression levels across all temporal parameters (1, 3, and 6 hours post-inoculation).
In the revised manuscript, we have incorporated expression level plots for four genes from S. aureus-inoculated samples and four genes from S. epidermidis-inoculated samples using maSigPro. Additionally, we have delineated supplementary quantitative parameters, including p-values, AUC measurements, and expression threshold values to strengthen the statistical foundation of our findings.
It should be noted that S. aureus-inoculated samples generated 2,439 statistically significant genes, rendering a comprehensive graphical analysis of the entire gene set impractical. Consequently, we prioritized our analytical focus on the four genes demonstrating superior statistical significance based on p-values and R-squared values, which represent the most promising candidates for subsequent investigation.
Encouragingly, our analysis identified the top four genes that exhibited both significant p-values and robust R-square values, and these genes consistently maintained elevated expression levels across all time points. These findings suggest that the observed gene expression profiles are not only statistically robust but may also have critical biological implications. For further clarification, a corresponding figure detailing these results has been incorporated into the revised manuscript.
Reviewer 2 Report
Comments and Suggestions for Authors
Abstract.
The summary is well-written; however, the combination of AUROC might cause some confusion, especially since ROC and AUC are treated separately. Please clarify this section for precision.
Introduction.
The introduction effectively addresses the topic and clearly outlines the problem, the study's direction, and the hypothesis being tested.
Material and methods
Line 120 does not specify which Illumina platform was used or mention the coverage or depth of sequencing.
Results
Lines 162–178: What criteria are used to determine whether a gene is considered regulated?
It would be beneficial to present these results in a table format for easier comprehension of which genes are categorized as more or less regulated.
How many dimensions were utilized in the PCA analysis?
In Figure 2, please include the regression equations.
Overall, the article is well explained. One recommendation is to incorporate additional tables, as many markers were identified for both S. epidermidis and S. aureus. This would enhance the clarity and visualization of the data.
References
Approximately 52% of the references are less than five years old, which is commendable.
Please increase the resolution for better clarity for Figures 3 (A, B, and C).
Comments on the Quality of English LanguageNo
Author Response
Comments 1: The summary is well-written; however, the combination of AUROC might cause some confusion, especially since ROC and AUC are treated separately. Please clarify this section for precision.
Response 1: We have modified the manuscript to replace the term "AUROC" with "AUC" in line 28-29, as suggested by the reviewer, to standardize and clarify our reporting.
Comments 2: Line 120 does not specify which Illumina platform was used or mention the coverage or depth of sequencing.
Response 2: In RNA-sequencing analysis, sequencing coverage and depth were characterized by read counts. Recognizing our limitations of total read count reporting, we enhanced the data presentation by incorporating the mean of mapped read counts and mapped read percentage. To improve methodological clarity, we eliminated the total number of raw reads from our analysis.
As requested by the reviewers, we have now specified the precise Illumina platform utilized in our RNA-sequencing methodology to enhance the technical transparency of our study (line 120-121, line 181-186).
Comments 3: Lines 162–178: What criteria are used to determine whether a gene is considered regulated?
Response 3: We concur with your recommendation and have subsequently incorporated specific criteria for the differentiation between upregulated and downregulated genes in '2.3. Differential Gene Expression Analysis' section (line 144-147).
Comments 4: It would be beneficial to present these results in a table format for easier comprehension of which genes are categorized as more or less regulated.
Response 4: We have incorporated your recommendation by including tables(Table 1, Table 2) that display differentially expressed genes in samples inoculated with S. aureus and S. epidermidis, highlighting both up-regulated and down-regulated gene sets in each respective condition.
Comments 5: How many dimensions were utilized in the PCA analysis?
Response 5: Principal component analysis in this paper was conducted utilizing eight dimensions to effectively reduce the complexity of our high-dimensional dataset while preserving the most significant sources of variation. In Figures 1 and 2, we visualized the RNA-sequencing data by projecting the first two principal dimensions.
Comments 6: Overall, the article is well explained. One recommendation is to incorporate additional tables, as many markers were identified for both S. epidermidis and S. aureus. This would enhance the clarity and visualization of the data.
Response 6: In accordance with your recommendation, we have incorporated a comprehensive table delineating the commonly significant genes observed in both S. aureus and S. epidermidis inoculated samples. This comparative presentation of differentially expressed genes serves to enhance the interpretability of our findings and facilitates more efficient comprehension of the transcriptional responses shared between these two staphylococcal species.
Comments 7: Please increase the resolution for better clarity for Figures 3 (A, B, and C).
Response 7: Following your recommendation regarding the enhancement of Figure 3(now Figure 2 after revision), we have implemented a comprehensive revision to improve its visual clarity.
Comments 8: In Figure 2, please include the regression equations.
Response 8: We concur with this assessment. However, the perfect separation observed in several genes found in the S. aureus inoculated samples prevented regression convergence, thereby precluding the formulation of a regression equation.
Round 2
Reviewer 1 Report
Comments and Suggestions for Authors
The authors adequately addressed all of my inquiries, and I don't have any additional questions.